# An Enhanced Detection System of Drill Rod Bending Degree Based on Two-Dimensional Laser

**DOI:** 10.3390/s20020370

**Published:** 2020-01-09

**Authors:** Lingling Wu, Guojun Wen, Yudan Wang, Xinjian Xu

**Affiliations:** 1School of Mechanical and Electronic Information, China University of Geosciences, Wuhan 430074, China; 13951211537@163.com (L.W.); wangyodan@163.com (Y.W.); 2Hubei Intelligent Geological Equipment Engineering Technology Research Center, Wuhan 430074, China; 3Wuhan Moming Technology Co., Ltd., Wuhan 430074, China; xing_xuxinjian@163.com

**Keywords:** two-dimensional laser, drill rod, bending degree, ellipse fitting, least square method

## Abstract

Drill rod straightness has to be strictly controlled and the maximum bending degree detection needs to be used in the straightening process. The mechanical bending degree measurement depends on machinery instruments and workers’ experience, often with low efficiency and precision. While the optical inspection, as a non-contact detection method, with higher precision and lower installation accuracy requirements, is frequently applied in the online detection system. Based on this, an enhanced bending degree detection system for a drill rod is proposed in this paper. Compared to the existing detection system, the main progress is to use a two-dimensional laser to quickly obtain arc profile data and fit with ellipse. Segment inspection idea is also utilized is this system as the camera that could obtain the whole drill rod in one shot needs extremely high resolution and price. A specialized algorithm is designed to fit the cross-section shape and whole centerline displacement based on the least square method. Some laboratory tests are conducted to verify this detection system, findings of which are compared to manual measured results. The maximum bending degree error is 2.14 mm and the maximum position error is 8.21 mm, which are both within the tolerance of error. Those results show the feasibility and precision of this enhanced detection system.

## 1. Introduction

Bending deformation frequently occurs in drill rod production process, impacting the quality of the screw threads machined at the ends of drill rods [1]. In addition, bent drill rods severely rub the borehole wall when they rotate in it, not only sharply reducing the useful life of drill rod, but also deteriorating the borehole wall effect, even worse, it may lead to the failure of the whole project. Therefore, it is required for manufacturers to detect and control the bending degree of drill rods. In order to meet the requirement of automatic mass production, the detection system requires high efficiency, high precision and automatic measuring.

The traditional mechanical measurement method with a dial indicator clearly fails to meet the high efficiency and accuracy requirements. Therefore, more and more optical inspection methods are applied into the detection system, such as current stereo-vision or multi-vision systems. Li et al. [2] proposed a monocular stereo vision-based contouring error detection method for CNC (Computerized Numerical Control) machine tools. From the acquired sequence images, the three-dimensional trajectories of rotary axes were reconstructed. In literature [3], Tang et al. has achieved real-time detection of surface deformation and strain in recycled aggregate concrete-filled steel tubular columns via four-ocular vision, in which calibration and coordinate associations among the four cameras are exploited to write a mathematical model and algorithm for multi-ocular visual point cloud image stitching to reconstruct the 3D convex curved deformation and compute the true strain of the specimens. Chen et al. [4] has built a four-camera vision system to obtain the visual information of targets. In the multi-camera schemes the causes of unavoidable global calibration errors are discussed in detail. Zhang et al. [5] proposed a fast and precise 3D reconstruction method for complex pipeline inspection by applying eight cameras to form four pair of stereo binocular cameras, obtaining the structural parameters. As for long shafts, the vision methods for deformation detection are not economically.

Except for stereo-vision, other vision systems like laser vision or machine vision are also widely employed in detection field. Sun et al. [6] has developed a measurement system by laser vision, in which laser sensors with CCD (charge-coupled device) cameras obtain light incision centers by image processing. Then pipe centerline is fitted with those centers. In this method, the cost increases with the length of pipes. Wu et al. [7] has tried using a one area-array CCD camera to shoot the whole long shaft in its rotating and then obtain the bending degree through image stitching and image processing. This method requires extremely high resolution of camera. Rana [8] has put forward a non-contact method for rod straightness measurement based on a quadrant laser sensor, but the ranges of the length and the bent of the measured rod were limited. Feng [9] has employed a single-mode fiber-coupled laser module for straightness measurement to enlarge the measuring range. Xiao et al. [10] realized the real-time measurement of straightness and roundness of spun workpieces during spinning by obtaining real-time images based on machine vision technology. Chu et al. [11] proposed a digital image-based method for the overall structural deformation monitoring, utilizing the image perspective transformation and edge detection. Qiu et al. [12] reported a novel method based on laser and machine vision to automatically measure tunnel deformation of multiple interest points in real time and effectively compensate for the environment vibration. Cui et al. [13] developed a tunnel deformation detection system called the railway mobile measurement system (RMMS) based on mobile laser scanning technique. In literature [14] Lu et al. proposed a machine vision-based smartphone LCD (liquid crystal display) surface deformation measurement method through applying the designed optical sensing system and a corresponding inspection algorithm.

There are still some other measurements based on kinds of sensors widely used in deformation measurement as well. Minoru [15] has presented a transmission-type position sensor for the straightness measurement, but it is only appropriate for large structure. In some other researches, pipe bending degree is evaluated by the cross-sectional eccentricity. Schalk [16] has measured the pipe eccentricity using laser triangulation. However, this method is error-prone for the laser collimator technique has been difficult to guarantee the measurement precision. Wang et al. [17] has proposed a contact measuring method for converting the eccentricity into the contacted lever fluctuation, and then the fluctuation is recorded by a displacement sensor. Nevertheless, it lacks marking out the bending position.

Based on the above, an enhanced drill rod bending degree detection system based on a two-dimensional laser is proposed in this paper considering both cost and precision. In this detection system, a laser sensor is driven to move along the drill rod so that the whole drill rod can be scanned by one sensor. Two-dimensional laser is intermittently projected onto drill rod profile, forming arc at different cross sections. Then these cross sections are fitted directly by those arc profile data, rather than the circle data spliced by numbers arcs, which avoids a drill rod being turned and guarantees the coplanarity of fitted data, largely increasing efficiency and accuracy. In addition, in the cross-section shape fitting, ellipse is adopted as the fitting model, making the fitting more realistic. It reduces parallel requirements of the driven system to the drill rod in the installation. After shape fitting, the cross-sectional centers are calculated based on the principle of plane geometry. In the center displacements calculation, a reference line is presupposed to change the spatial curve displacement into plane distances between centers to one line, greatly increasing the program computation speed. Finally, the integrated centerline displacement is fitted by a polynomial with four orders due to its fast computing rapid. Laboratory tests were done to verify the system precision and reliability.

## 2. Detection Principle

Drill rod bending degree means the bending degree of its centerline and it can be characterized by some parameters such as curvature, eccentricity, displacement of centerline, etc. [18]. Since the calculation of curvature and eccentricity of a spatial curve has been numerically too demanding, centerline displacement is often employed to characterize the bending degree in real-time detection. As one centerline consists of infinitely many continuous cross-sectional centers that is impossible to be measured one by one. What is more, the drill rods are long shafts. Consequently, picking several cross sections at regular intervals, finding their center displacements, and then fitting the whole centerline displacement become an effective way to evaluate the drill rod bending degree.

As shown in Figure 1, a laser sensor projects two-dimensional laser onto surface of drill rod to form diffuse reflection. Some reflected light comes back to the sensor and is imaged on the complementary metal-oxide-semiconductor (CMOS), the arc profile is measured by detecting changes in shape. Then the laser sensor could move along with the drill rod, be triggered to scan more cross-sectional profile, further calculate center displacements and fit the whole drill rod bending degree curve.

### 2.1. Cross-Section Shape Fitting

As shown in Figure 2, an arc is formed on drill rod profile, the distances between each laser spots (in the arc) and the sensor (namely the *z*-coordinates) can be measured directly by the sensor. To further illustrate the laser scanning and ellipse fitting, a Cartesian coordinate system is established, where the origin of the coordinate P is the position of the laser sensor, *Z*-axis is the direction from the sensor to the drill rod, and *X*-axis is perpendicular to *Z*-axis at P in the cross-section plane. In order to greatly simplify the installation of drill rod, in the shape fitting, ellipse is employed as the fitting model for the light incision is hardly completely perpendicular to the drill rod longitudinal axis in such installation, as a result, the intersecting cross section is barely circular, but must be an ellipse.

As those laser spots are evenly distributed in the two-dimensional laser, so their *x*-coordinates are easily obtained, while their *z*-coordinates are directly measured by the sensor. Therefore, the laser spot coordinates can be achieved as
(1)$$\;data=\left(\begin{array}{cc} x_{1} & z_{1}\\ x_{2} & z_{2}\\ \cdot & \cdot \\ \cdot & \cdot \\ \cdot & \cdot \\ x_{n} & z_{n} \end{array}\right).$$

In the shape fitting, the least square method fitting of an ellipse is selected.

The algebraic distance of a point to an ellipse is,
(2)$$P_{1}x^{2}+P_{2}z^{2}+P_{3}xz+P_{4}x+P_{5}z+P_{6}=0,$$
where $P_{1}$, $P_{2}$, $P_{3}$, $P_{4}$, $P_{5}$ and $P_{6}$ are equation parameters and x and z are coordinates. There could be n such equations when given n coordinates, which can be formulated as,
(3)$$\left(\begin{array}{cccccc} x_{1}{}^{2} & z_{1}{}^{2} & x_{1}z_{1} & x_{1} & z_{1} & 1\\ x_{2}{}^{2} & z_{2}{}^{2} & x_{2}z_{2} & x_{2} & z_{2} & 1\\ \vdots & \vdots & \vdots & \vdots & \vdots & \vdots \\ x_{n}{}^{2} & z_{n}{}^{2} & x_{n}z_{n} & x_{n} & z_{n} & 1 \end{array}\right)\left(\begin{array}{c} P_{1}\\ P_{2}\\ \vdots \\ P_{6} \end{array}\right)=0.$$

Ellipse fitting is the course to evaluate those parameters according to these equations.

### 2.2. Cross-Sectional Center Calculation

After the ellipse is fitted out, the center coordinate can be calculated based on the plane geometric theory. As shown in Figure 3, an ellipse in *XOZ* plane, its center can be noted as $O^{\prime }(x_{0},z_{0})$, two lines paralleled to *Z*-axis are tangent to the ellipse, the ellipse center is the midpoint of these two tangency points as ellipse is symmetrical. Assuming the tangent line is
(4)x=b.

Putting Equation (4) into Equation (2), a new formulation is achieved as following
(5)$$p_{2}z^{2}+(p_{3}b+p_{5})z+p_{1}b^{2}+p_{4}b+p_{6}=0.$$

As the lines are tangent to the ellipse, we can derive the discriminant Δ from Equation (5) and,
(6)$$\Delta =(P_{3}{}^{2}-4P_{1}P_{2})b^{2}+(2P_{3}P_{5}-4P_{2}P_{4})b+(P_{5}{}^{2}-4P_{2}P_{6})=0.$$

Further,
(7)$$b_{1}+b_{2}=\frac{4P_{2}P_{4}-2P_{3}P_{5}}{P_{3}{}^{2}-4P_{1}P_{2}}.$$

Therefore, *x*-coordinate value of center is
(8)$$x_{0}=\frac{b_{1}+b_{2}}{2}=\frac{2P_{2}P_{4}-P_{3}P_{5}}{P_{3}^{2}-4P_{1}P_{2}}.$$

As well,
(9)$$z_{0}=\frac{2P_{1}P_{5}-P_{3}P_{4}}{P_{3}^{2}-4P_{1}P_{2}}.$$

### 2.3. Centerline Displacement Calculation and Fitting

In drill rod straightening, to avoid re-bending, workers need to know the maximum bending degree and the position to push the deformation back by reverse pressure. During that process, drill rod ends are clamped by holders. Therefore, the straight line (dashed line in Figure 4) connecting these two ends is set as criteria to evaluate center displacement, we may call it the reference line as well. After that, the distances between the reference line with the centers can be calculated as center displacements.

In the procedure, a new coordinate system is established as shown in Figure 4. The light incisions are in the *XOZ* plane, and *Y*-axis is parallel to the drill rod longitudinal axis. The first scanned and last scanned cross sections are picked as the two ends. The straight line connecting their centers is
(10)$$\frac{x-x_{1}}{x_{1}-x_{n}}=\frac{y-y_{1}}{y_{1}-y_{n}}=\frac{z-z_{1}}{z_{1}-z_{n}},$$
where $\left(x_{1},y_{1},z_{1}\right)$ are coordinates of the first scanned cross-sectional center and $\left(x_{n},y_{n},z_{n}\right)$ are coordinates of the last scanned cross-sectional center. According to the vector product, distance from center $\left(x_{i},y_{i},z_{i}\right)$ to line is
(11)$$d_{i}=\frac{\sqrt{\left|\begin{array}{cc} x_{i}-x_{1} & y_{i}-y_{1}\\ x_{1}-x_{n} & y_{1}-y_{n} \end{array}\right|+\left|\begin{array}{cc} y_{i}-y_{1} & z_{i}-z_{1}\\ y_{1}-y_{n} & z_{1}-z_{n} \end{array}\right|+\left|\begin{array}{cc} z_{i}-z_{1} & x_{i}-x_{1}\\ y_{1}-y_{n} & x_{1}-x_{n} \end{array}\right|}}{\sqrt{{\left(x_{1}-x_{n}\right)}^{2}+{\left(y_{1}-y_{n}\right)}^{2}+{\left(z_{1}-z_{n}\right)}^{2}}},$$
where $\left(x_{1},y_{1},z_{1}\right)$ are the first scanned cross-sectional center; $\left(x_{n},y_{n},z_{n}\right)$ are coordinates of the last scanned cross-sectional center and $\left(x_{i},y_{i},z_{i}\right)$ are coordinates of the i−th cross-sectional center.

After computation all the distances, the data are
(12)$$d={\left(d_{1},d_{2},\cdot \cdot \cdot ,d_{j}\right)}^{T},$$
characterizing center displacements, in which j=n−2. In this way, spatial curve bending calculation is converted into the distances between points and a line in two dimensions, which could greatly improve the speed of the algorithm.

In order to find the position of maximum bending degree of the whole drill rod, longitudinal axis curve fitting is applied on the discrete center displacements and their *y*-coordinates based on least square method. The data under fitting are
(13)$$d_{f}=\left(\begin{array}{cc} y_{1} & d_{1}\\ y_{2} & d_{2}\\ \cdot & \cdot \\ \cdot & \cdot \\ \cdot & \cdot \\ y_{j} & d_{j} \end{array}\right),$$
where j is determined in the previous process, and j=n−2, $y_{j}$ is the *y*-coordinate corresponding to the cross section and $d_{j}$ is the center displacement. Among numbers of curve fitting methods, a polynomial fitting can be performed because of its speedability and illustrating of the center displacement along with the longitudinal axis. Therefore, the maximum value of the fitted curve refers to the maximum bending degree and its abscissa refers the position the maximum bending degree occurs, namely the places that need to be straightened.

## 3. Detection System Design

According to the detection principle, a drive system is designed to bring the laser sensor to move along with the drill rod. In this case, the laser sensor need not keep scanning the drill rod, but be triggered at the designated position. The detection system model is shown in Figure 5. For the laser sensor as a non-contact measuring method, the system installation is greatly simplified. The drill rod (1) under test is put on two V-blocks (2), with a distance away from the laser sensor. The leading screw (5), supported on bearings (4), is driven by the motor (3) through the coupler (not shown). The laser sensor (7) is fixed on the screw nut (6), which is mounted on a guide rail (not shown), so that it moves along with the drill rod. The whole drive system is arranged on a support frame (8). It is better to install the drill rod closely to the laser sensor horizontal alignment to obtain as much measuring data as possible, which could improve the detection precision.

As the laser sensor is driven by the driven system, and it is the origin of the coordinate in shape fitting, so the straightness error of the driven system directly influences the cross-sectional center coordinate values, further decreases the veracity of the detection system. In this system, the driven system consists of a nut screw pair for its high transmission efficiency, good stiffness and high transmission precision matched with a servo motor, and a guide rail to drive the sensor to smoothly move along with the drill rod. Therefore, in order to reduce the influence, the high straightness accuracy of the driven system is highly required. In order to improve the hardware reliability and to prevent the screw nut impacting bearings, two limit switches (9; shown only one) are respectively installed on the rack near these two bearings. If the screw nut reaches the limit position, the limit switch immediately sends signals to the personal computer (PC), which then, controls the leading screw to stop rotating or reverse at once. The installation position of two limit switches also limits the maximum range of laser sensor movement.

A specialized algorithm is developed to control the whole detection system to acquire data and process data, whose block diagram is shown in Figure 6. In detail, data acquisition contains device connection and parameter setting. Data processing mainly includes ellipse fitting, center calculation, centerline displacement fitting and result display. As the drive program of the laser sensor had been developed on MFC (Microsoft Foundation Classes), so the data acquisition algorithm has to be compiled with C based on MFC; while data processing and displaying is designed in MATLAB R2014a for its powerful data processing functions. Then the data processing part is embedded into MFC to make an integrated detection software system.

### 3.1. Data Acquisition

Data acquisition is to control the whole hardware system via signal transmission to scan the profile of drill rod and obtain data. The control module is detailed as Figure 7. The laser sensor is driven by its self-contained controller, which communicates with the PC through USB. The servosystem and microcontroller, connected with PC through serial port, respectively control the servo motor and the limit switches. As said before, the laser sensor scans the profile just when it is triggered.

The key of this module is how to trigger the laser sensor to start to illuminate two-dimensional laser. There are two methods: one is time trigger and the other is displacement trigger. In this system, displacement trigger is adopted for the sample interval is more concerned about and more intuitional. In order to trigger the laser sensor by displacement, its controller should also connect with the encoder of servosystem to further get Z-phase signal, which will plus one if the leading screw turns one circle, namely, the Z-phase signal is proportional to the number of turns of the leading screw. In this way, the sample interval can be transformed into the turns of the leading screw combined with its helical pitch.

For making the data acquisition part easy, an interface (Figure 8) is designed to connect device and set parameters. Before scanning, PC needs to establish connection with servosystem and limitation system by choosing the right serial port in the drop-down box. In addition, the motor speed and the sample interval are directly input in the box. Scanned data is saved simultaneously in a defaulted file unless the data save path is user-defined. After determining the direction of screw rotation, the devices could work to acquire data.

### 3.2. Data Processing

There are 800 laser spots in the two-dimensional laser so that in one scanning we could get 800 data immediately, namely 800 *z*-coordinates. Those laser spots distribute evenly from −40.0 to +39.9 mm, with 0.1 mm space between every two laser spots, so *x*-coordinates can be easily calculated. For so long drill rod that many cross sections need to be scanned. In order to distinguish data of different cross section, every cross-section data are set to be saved as one row matrix in a same file. In later data processing, *x*-coordinates have to correspond to *z*-coordinates one by one. Thus, *x*-coordinates are listed as the first row in the same file. When it is needed to find the paired *x* coordinates, the program will traverse the first row. Therefore, after completing scanning one drill rod, a large matrix of scanned data is obtained, the style of the data matrix is shown as below,
(14)$$\left(\begin{array}{ccccccccc} -40.0 & -39.9 & -39.8 & \cdot \cdot \cdot & 0 & \cdot \cdot \cdot & 39.7 & 39.8 & 39.9\\ Z_{1,1} & Z_{1,2} & Z_{1,3} & \cdot \cdot \cdot & Z_{1,401} & \cdot \cdot \cdot & Z_{1,798} & Z_{1,799} & Z_{1,800}\\ Z_{2,1} & Z_{2,2} & Z_{2,3} & \cdot \cdot \cdot & Z_{2,401} & \cdot \cdot \cdot & Z_{2,798} & Z_{2,799} & Z_{2,800}\\ Z_{3,1} & Z_{3,2} & Z_{3,3} & \cdot \cdot \cdot & Z_{3,401} & \cdot \cdot \cdot & Z_{3,798} & Z_{3,799} & Z_{3,800}\\ \vdots & \vdots & \vdots & \ddots & \vdots & \ddots & \vdots & \vdots & \vdots \\ Z_{n,1} & Z_{n,2} & Z_{n,3} & \cdot \cdot \cdot & Z_{n,401} & \cdot \cdot \cdot & Z_{n,798} & Z_{n,799} & Z_{n,800} \end{array}\right),$$
where the first subscript of Z means the number of cross-section, the second one means the number of data. The second row is the data of the first cross-sectional profile, the third row is the second cross sectional profile data, the last row is data of the *n*-th cross section data and so on; each cross section has 800 data.

The procedure in MATLAB reads the scanned data, so it loads in the large (n+1)∗800 matrix, which contains lots of invalid data, especially at the ends of row matrixes. That is because a two-dimensional laser is longer than the diameter of the drill rod, so laser ends fail projecting on the drill rod. In the middle of matrix, some abnormal data exists because of bad surface quality. Therefore, finding those data and filtering them is essential at the beginning of processing data. After observing multiple sets of scanned data, we find that the invalid data at the ends of matrix are always extremely small, which can be easily distinguished from the valid data. The abnormal data can be recognized by differentiating it with the data beside. If their different value exceeds a given threshold, this data will be deleted. It is worth noting that once the invalid *z*-coordinates are identified, their positions in the matrix need also be discerned by the functions, therefore, if they are deleted, their paired *x*-coordinates are deleted as well.

Take the first cross section data as an example, after filtering, the second row matrix in the large matrix $\left(\begin{array}{ccccccccc} Z_{1,1} & Z_{1,2} & Z_{1,3} & \ldots & Z_{1,401} & \ldots & Z_{1,798} & Z_{1,799} & Z_{1,800} \end{array}\right)$ becomes a new row matrix $\left(\begin{array}{cccc} Z_{1,i} & Z_{1,i+1} & \ldots & Z_{1,i+j} \end{array}\right)$. Then they needed to be converted into an independent matrix with paired x- and z-coordinates, as shown following,
(15)$$\left(\begin{array}{cc} X_{i} & Z_{1,i}\\ X_{i+1} & Z_{1,i+1}\\ \vdots & \vdots \\ X_{i+j} & Z_{1,i+j} \end{array}\right).$$

This independent matrix means the coordinate pairs of useful laser spots in each cross section, which will be used in Equations (2) and (3) to fit the ellipse of that cross section. Then the cross-sectional center and center displacement are calculated according the detection principle. At last, the centerline maximum bending degree is calculated and its position is found out.

## 4. Laboratory Test

In order to verify the feasibility and veracity of the detection system, some tests were implemented in the laboratory. As shown in Figure 9, the whole detection system was established according to the model. As the straightness of the driven system should be strictly controlled, the lead screw was customized for such long length with high straightness. Additionally, the guide rail was bought as a ultra-high-precision rail, with the straightness accuracy within 1 μm/100 mm. The laser sensor was fixed on the nut to avoid swing in scanning and moving. The testing drill rod was a trenchless one, with a maximum bending degree of 5.4 mm positioned in its 152-cm, which was measured by the mechanical method. It had an inner diameter of 65 mm, an outer diameter of 75 mm and a length of 2700 mm, including the screw threads at the ends. In the process of physical installation, the most need to pay attention to is that the guide rail should be highly parallel to the leading screw to reduce frictional resistance and improve transmission accuracy, as for the drill rod, it is not necessary to be put completely paralleled to the leading screw.

After the laser sensor, limit switches and motor successfully connected with the PC, the motor speed and the sample distance were set in the data acquisition interface. When the program worked, the laser sensor first returned to the first end of drill rod (approaching the first limit switch), and then moved along the drill rod. While the laser sensor moved the sample interval, it emitted a two-dimensional laser on the profile and obtained data of the cross-sectional profile. Since the scanning time was extremely short, the sensor will not stop moving when scanning the drill rod. After the data acquisition completed, all data will be saved as a large matrix in a .txt file, which will be read directly by the data processing program in the MATLAB and be processed automatically, finally the results of it will be displayed in the given text box on the interface (Figure 10).

As said before, the data scanned by laser sensor is a huge matrix with (*n* + 1) rows and 800 lines, as part of it is shown in Figure 10 (data are copied from .txt file into excel to show clearly). The first row matrix represent the spacing of laser points, namely *x*-coordinates, whose values could be calculated by (−40+p∗0.1) as *p* means the column number. Then the following row matrixes were scanned data of cross sections, namely *z*-coordinates. As we can see, the invalid data in the red box were extremely small, and were easily distinguished with other valid data. With the help of functions in MATLAB, the valid data of each cross section could be found and saved in a vector *a*1, as well, the column number *p* needs be also found to calculate *x*-coordinates saved in a vector *a*2. Combined with these two row vectors *a*1 and *a*2, the paired *x*- and *z*-coordinates of valid laser spots were formed and could be used to fit the shape of the cross section. However, the data achieved by the laser sensor were down to four decimal places, and the number of valid data of one cross section after filtering was still more than 200. It is hard to employ all data to fit the ellipse efficiently. Therefore, in the algorithm, the data used to fit were picked out in equal space, picking the first datum in 20 data so that we could get about 10 data to fit the ellipse of the cross section.

In the laboratory test, five groups of experiments with different sample intervals of 10 cm, 16 cm, 24 cm, 28 cm and 34 cm were conducted on the same drill rod. Theoretically, the shorter the sample distance is, the higher the detection accuracy is obtained. To make the test more persuasive, when the sample intervals was 16 cm and 24 cm, the drill rod was turned with 180 degree so that the scanned arc profile was on the other side with other three group experiments. Table 1 shows the calculated maximum bending degree and its position in different sample intervals. The maximum bending degree ranged from 5.15 to 7.54 mm, and maximum bending deformation positions between 143.79 and 153.41 cm. Compared with the mechanical measuring result, the maximal error of the maximum bending degree was about 2.14 mm, and it happened when the sample distance was 34 cm; the maximal error of maximum bending deformation position was about 8.21 cm, this appeared when the sample distance was 34 cm, too. For the detection system, the differences controlled within 2% of the length could be applied in real detection. Therefore, for the 2700-mm length drill rod, only the results at 34 cm sample distance could not meet the detection system accuracy requirement. This also reminds us that the sample distance should not be so large that the valid fitted cross-section become less, which will fail meeting the detection accuracy. Even though, the results at five different sample intervals are still relatively closes.

Figure 11 shows the results when the sample distance was 24 cm. This result display interface contains two parts, one is to show the ellipse fitting of each cross section; the other is to show the fitted curve of the whole drill rod. The ellipse fitting part was designed to improve program visibility, as every cross-section fitting could be searched here and checked. If the ellipse fitting or the used data is unsatisfactory, this part display could help manufacturers to identify the improper fitting. Else, this part can be ignored. Above the curve of the drill rod, a text box is set here to show the concrete maximum bending degree and its position, making it an effective way to get the result. In Figure 11, the “Ellipse Fitting” shows the fifth cross section ellipse fitting. In the text box are center coordinates and in the axis is the ellipse after fitting. From the fifth fitting picture, we can clearly find that the cross-sectional profile was an ellipse, not a circle. The “Bending Degree” shows the maximum bending degree and its position in the text box and the whole displacement curve in the axis. The maximum bending degree and its position were 5.86 mm and 153 cm respectively, which shows this drill rod needs be straightened at the position.

## 5. Conclusions

The bent drill rod has many disadvantages in transforming torque, especially in the deep drilling process. When bent drill rods rotate in the borehole, they will scrape the borehole wall. This not only deteriorates the wall effect, but also reduces the drill rod lifetime and the transmission efficiency. Therefore, manufacturers are required to provide drill rods with high straightness. They need to detect bending degree of the drill rod and straighten those bent drill rods before selling them to purchasers.

Currently, the detection system is gradually getting rid of mechanical instruments with manual observation and increasingly using optical inspection technology. An enhanced new detection system was proposed in this paper, which bases on two-dimensional laser measuring technique. By this system, it is not necessary to turn the testing drill rod, since almost semicircle measuring data are adequate to fit out this cross section. As a laser sensor moves along with the drill rod, more and more cross section data could be collected. After filtering the original data, every cross-sectional profile could be fitted out using ellipse model based on the least square method. Further, centers of each cross section could be calculated using those ellipses based on geometry. After choosing the line connecting the first and the last ellipse centers as a reference line, those discrete center displacements could be evaluated. Considering the place scanned may not be the maximum bending degree positioned, so those discrete center displacements are used to fit out a curve representing the centerline displacement, therefore its maximum value and abscissa mean the maximum bending degree and position.

In order to verify this proposed detection system tests were conducted on a trenchless drill rod in our lab. The detection results in different sample intervals were close, and highly consistent with the results of manual measurement. From the results of 24 cm sample distance test, the center displacement in the second cross section was the biggest among all, but after they were fitted, the maximum of this fitted curve was near the fifth cross section, which was consistent with the fact. This shows the necessity of curve fitting. In those five tests, the maximal maximum bending degree error was 2.14 mm and maximal position error was 8.21 cm. What is more, in the laboratory test, it only took 14 s to give out the detection result of one drill rod. Of course, the detection time could be much less if we set the motor much faster. These tests showed the feasibility and reliability of the proposed system.

Above all, this system was able to calculate and position the maximum bending degree of a drill rod quickly and accurately, meeting requirements of mass production. It improved the automatic lever of a detection system and reduced workers labor intensity. However, for the length of the two-dimensional laser, this system can only detect those shafts whose diameters are within the laser length, else the detection precision will decrease.

## 6. Patents

The patent “Drill pipe bending degree detection device and method” is resulting from the work reported in this manuscript.

## Figures and Tables

**Figure 1 sensors-20-00370-f001:**
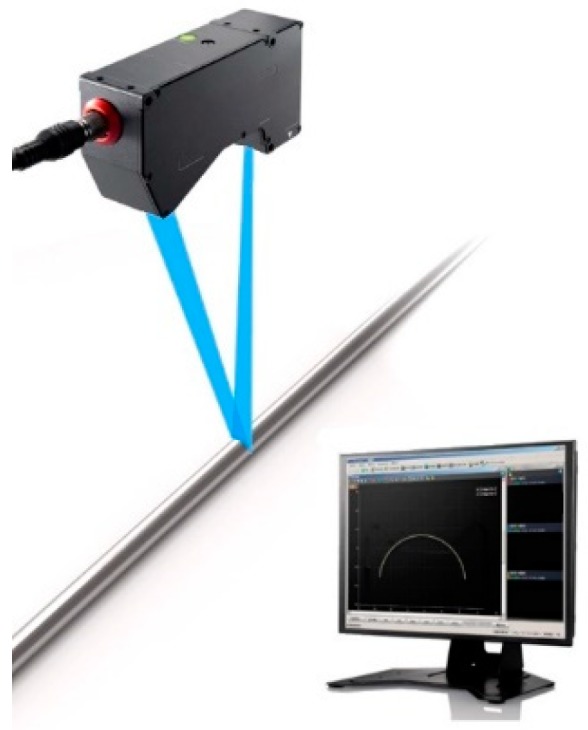
Process of laser scanning drill rod.

**Figure 2 sensors-20-00370-f002:**
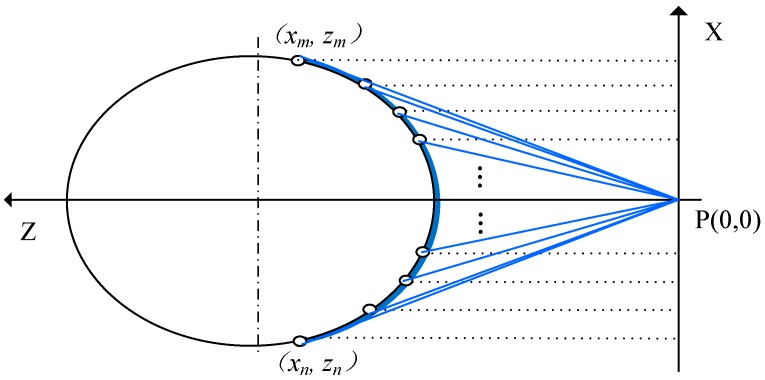
Schematic diagram of laser spots distribution in a Cartesian coordinate system.

**Figure 3 sensors-20-00370-f003:**
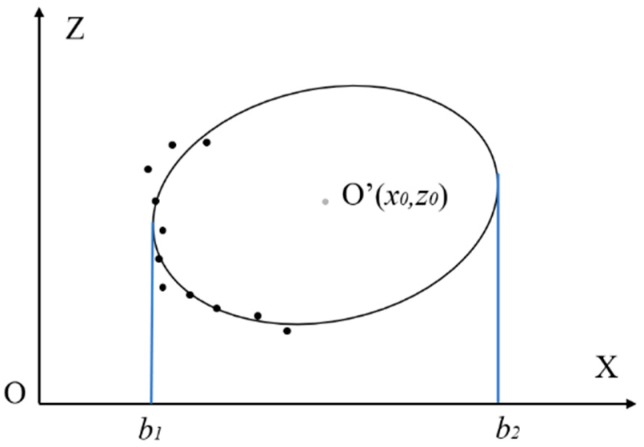
Ellipse center calculating schematic.

**Figure 4 sensors-20-00370-f004:**
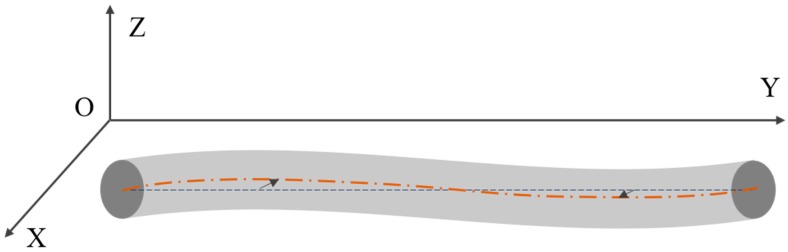
The displacement between reference line and centerline.

**Figure 5 sensors-20-00370-f005:**
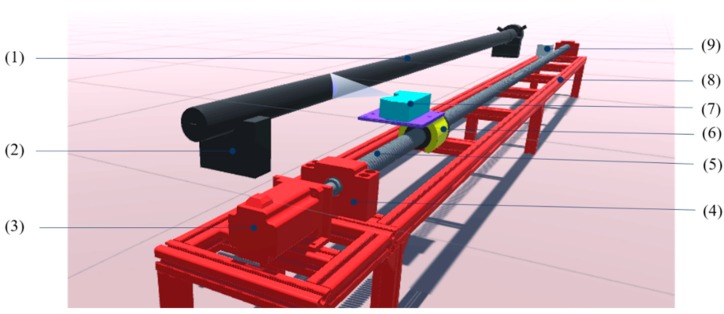
The model of detection system. (1) Drill rod, (2) V-block, (3) motor, (4) bearing, (5) leading screw, (6) screw nut, (7) laser sensor, (8) support frame and (9) limit switch.

**Figure 6 sensors-20-00370-f006:**
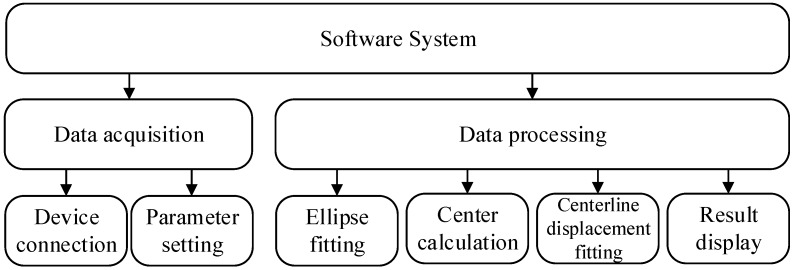
Software design block diagrams.

**Figure 7 sensors-20-00370-f007:**
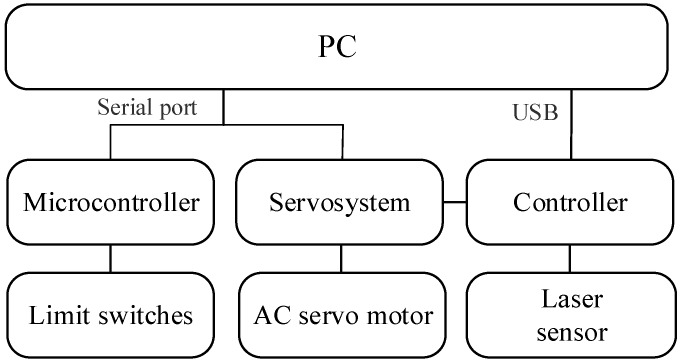
Control module block diagrams.

**Figure 8 sensors-20-00370-f008:**
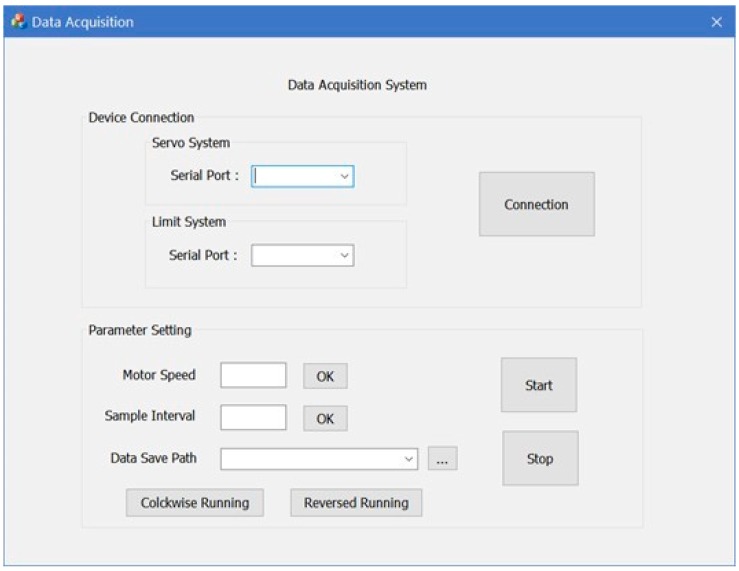
Data acquisition interface.

**Figure 9 sensors-20-00370-f009:**
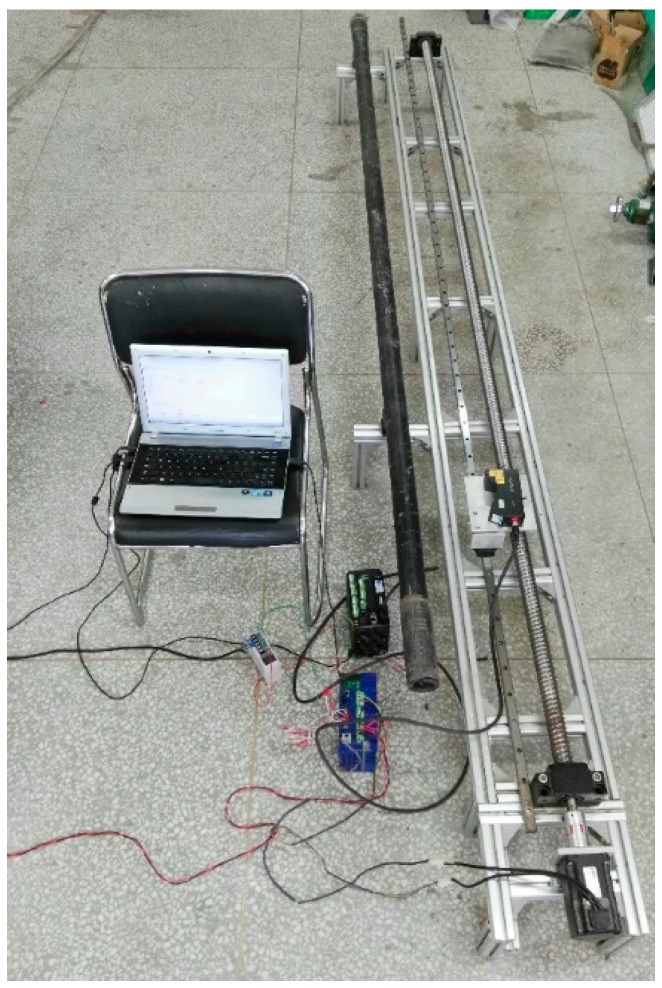
Physical map of the detection system.

**Figure 10 sensors-20-00370-f010:**
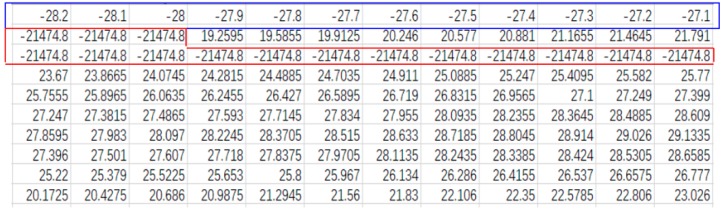
Part of scanned data with 24 cm intervals in excel.

**Figure 11 sensors-20-00370-f011:**
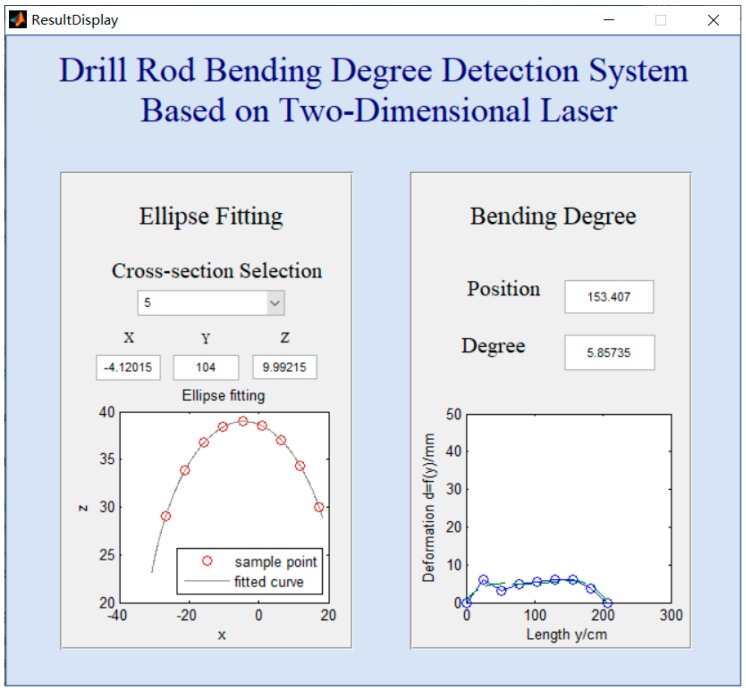
Result of 24 cm sample interval.

**Table 1 sensors-20-00370-t001:** The maximum bending degree with its position in different sample intervals.

Sample Interval (cm)	Maximum Bending Degree (mm)	Maximum Bending Position (cm)
10	5.14523	152.113
16	6.74561	147.374
24	5.85735	153.407
28	5.43628	149.244
34	7.54299	143.789

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
