# Peer review of "An Enhanced Detection System of Drill Rod Bending Degree Based on Two-Dimensional Laser"

_sensors, 2020, doi:10.3390/s20020370_

Round 1

Reviewer 1 Report

    In this manuscript, authors built a simple system, and use two-dimensional laser to scan the drill rod, fit cross section based on least square method, and finally get the centerline displacement of the rod. Some laboratory tests are conducted to verify this detection system. Although the measurement accuracy is not so high, it may work. So, I recommend that manuscript be published in Sensors after some modifications.

Authors should clearly put out the accuracy requirements for measuring the drill rod, so readers can just if the method can be applied in the real drill rod measurements. As the laser head is driven along a long guide, clearly the straightness error of the guide influences the measurement accuracy, the authors should discuss it, and put out how to reduce the influence.

Reviewer 2 Report

The authors propose an enhanced bending degree detection system for drill rod, which has higher detection accuracy and efficiency than manual detection. The detection method proposed in this manuscript is relatively novel and practical. There are suggestions for improving the quality of manuscripts.

In the process of verifying this detection system, only five tests were used. Please clarify the consideration. Add more up-to-date references in the introduction. Steoro-vision method can realize non-contact detection and 3D reconstruction for the surface deformation/crack recognition of the infrastructures. It is another sharp tool of the structural health monitoring box as well as non-destructive technology including laser sensors. The authors may introduce the vision method in-depth for the integrity of the introduction. References are encouraged citing below (High-accuracy multi-camera reconstruction enhanced by adaptive point cloud correction algorithm, Real-time detection of surface deformation and strain in recycled aggregate concrete-filled steel tubular columns via four-ocular vision). Please put the publication information and other citations of citations 5 and 6 in a uniform format. Do not blank at the beginning of line 242. The number in the input box in Figure 10 (Result of 24cm sample interval) is not very clear. It is recommended to enlarge the picture. Figure 10 only shows the results of the 24 cm sampling interval. To be more convincing, please display the results of the other four lengths of the sampling interval. Illumination has an impact on the accuracy of the camera. Please explain the lighting conditions in the experimental environment and whether it has an impact on the experiment. The paper took a lot of pen and ink to introduce the principle and process of measurement. However, the experiment and computer configuration are missing.

Reviewer 3 Report

       This manuscript presents an enhanced detection system to determine the degree of bent in a drill rod using 2D laser. The system is used to optically scan the drill rod and determine its cross-section by fitting it with an ellipse using least squares-based regression. Although the problem is interesting the advancement reported in this manuscript by itself is not substantial enough. Further, the detail regarding how this new system performs against the existing techniques in this field needs to be addressed. Professional English editing is recommended. The least square analysis, the code use etc. are detailed elaborately, which may not be essential. While, the detail surrounding the sensor itself and what is novel in this new system needs to be emphasized. Also, the details around the translation and scanning system etc should be included.

Round 2

Reviewer 2 Report

The manuscript can be accepted after revision.

Reviewer 3 Report

Accept